# DURENDAL: GRAPH DEEP LEARNING FRAMEWORK FOR TEMPORAL HETEROGENEOUS NETWORKS

## ABSTRACT

Temporal heterogeneous networks (THNs) are evolving networks that characterize many real-world applications such as citation and events networks, recommender systems, and knowledge graphs. Although different Graph Neural Networks (GNNs) have been successfully applied to dynamic graphs, most of them only support homogeneous graphs or suffer from model design heavily influenced by specific THNs prediction tasks. Furthermore, there is a lack of temporal heterogeneous networked data in current standard graph benchmark datasets. Hence, in this work, we propose DURENDAL, a graph deep learning framework for THNs. DURENDAL can help to easily repurpose any heterogeneous graph learning model to evolving networks by combining design principles from snapshot-based and multirelational message-passing graph learning models. We introduce two different schemes to update embedding representations for THNs, discussing the strengths and weaknesses of both strategies. We also extend the set of benchmarks for TNHs by introducing two novel high-resolution temporal heterogeneous graph datasets derived from an emerging Web3 platform and a well-established e-commerce website. Overall, we conducted the experimental evaluation of the framework over four temporal heterogeneous network datasets on future link prediction tasks in an evaluation setting that takes into account the evolving nature of the data. Experiments show the prediction power of DURENDAL compared to current solutions for evolving and dynamic graphs, and the effectiveness of its model design.

## 1 INTRODUCTION

Graph neural networks (GNNs), as a powerful graph representation technique based on deep learning, have been successfully applied to many real-world static and heterogeneous graphs (Bing et al., 2022). Recently, GNNs also attracted considerable research interest to learn, extract, and predict from evolving networks, which characterize many application domains, such as recommender systems (You et al., 2019), temporal knowledge graphs (Cai et al., 2022), or social network analysis (Dileo et al., 2022). However, the success of heterogeneous graph learning has not entirely transferred to temporal heterogeneous networks (THNs).

Current architectural designs for dynamic GNNs have been proposed for homogeneous graphs only. A few heterogeneous graph learning models try to extend the computation to handle the graphs' dynamic but suffer limitations in model design, evaluation, and training strategies. Specifically, they struggle to incorporate state-of-the-art designs from static GNNs, limiting their performance. Their evaluation settings are fixed train test splits, which do not fully reflect the evolving nature of the data, and commonly used training methods are not scalable. Furthermore, existing solutions for learning from THNs are heavily designed to solve a specific prediction task, i.e. knowledge base completion, making it hard to obtain general-purpose embedded representation for nodes, edges, and the whole graphs.

To overcome the limitations described above, we propose DURENDAL, a graph deep learning framework for temporal heterogeneous networks. Inspired by the ROLAND (You et al., 2022) framework for dynamic homogeneous graphs, DURENDAL can help to easily repurpose any heterogeneous graph learning model to dynamic graphs, including training strategies and evaluation settings for evolving data. The ability to easily extend heterogeneous GNNs to the dynamic setting arises from a combination of model design principles. To handle dynamic aspects we consider the node em-

beddings at different GNN layers as hierarchical node states, recurrently updating them over time through customizable embedding modules. Additionally, to handle the heterogeneity, we introduce heterogeneous hierarchical node states and customizable semantic aggregation schemes. In this way, modern architectural design options such as skip connections or attention mechanisms can be easily incorporated. We propose two different update schemes for temporal heterogeneous node states discussing their strengths and drawbacks in terms of scalability, memory footprint, and learning power, allowing researchers to easily follow one of the two schemes according to the real application scenario they face.

We train DURENDAL using an incremental training procedure and using a live-update setting for the evaluation. We conducted experiments over four different THNs network datasets on future link prediction tasks. The four datasets were selected based on certain minimum requirements that they had to meet in order to serve as useful testing grounds for temporal heterogeneous graph learning models. Since current graph benchmarks for THNs are very limited, we also extend the set of benchmarks for TNHs by introducing two novel high-resolution temporal heterogeneous graph datasets derived from an emerging Web3 platform and a well-established e-commerce website.

The experimental evaluation shows the prediction power of DURENDAL and the effectiveness of its model design and update schemes. DURENDAL achieves better performance compared to current solutions for dynamic graphs on three of the four datasets, which exhibit different time granularity, number of snapshots, and new incoming links. The effectiveness of the DURENDAL model design is shown by the increase in performance of state-of-the-art heterogeneous graph learning models repurposed in a dynamic setting with our framework, which also highlights the benefit of some modern architectural design options for GNNs. Lastly, we compare the two different DURENDAL update schemes with the ROLAND one, showing the improvements in the prediction performance of our schemes.

We summarize our main contributions as follows: *i)* we propose a novel graph deep learning framework that allows an easy repurposing of any heterogenous GNNs to a dynamic setting; *ii)* we introduce two different update schemes for obtaining temporal heterogeneous node embeddings, highlighting their strengths and weaknesses and their practical use scenarios; *iii)* we define some minimal requirements datasets must satisfy to be useful testing grounds for temporal heterogeneous graph learning models, extending the set of benchmarks for THNs by introducing two novel high-resolution THNs datasets; and *iv)* we evaluate different types of approaches for dealing with THNs in the new live-update setting, enabling an assessment of the performances along the snapshots of the evolving networks.

## 2 RELATED WORK

**Temporal GNNs.** GNNs have been successfully applied to extract, learn, and predict from temporal networks as surveyed in Longa et al. (2023). Most of the works combine GNNs with recurrent models (e.g. a GRU Cell (Chung et al., 2014)): adopting GNN as a feature encoder (Peng et al., 2020), replacing linear layers in the RNN cells with GNN layers (Zhao et al., 2020; Li et al., 2017; Seo et al., 2018), or using RNNs to update the learned weights (Pareja et al., 2020). Other works combine GNN layers with temporal encoders (Xu et al., 2020) or extend the message-passing computation on temporal neighborhood (Luo & Li, 2022; Zhou et al., 2022). All these works have been proposed only for homogeneous graphs. Moreover, most have limitations in model design, evaluation, and training strategies, as shown in You et al. (2022).

**Temporal Heterogenous GNNs.** Only a few works on heterogeneous graph deep learning try to extend the reasoning over temporal networks. For instance, Jin et al. (2020) and Li et al. (2021b) employ a recurrent event encoder to encode past facts and use a neighborhood aggregator to model the connection of facts at the same timestamp. Hu et al. (2020b), inspired by Transformer positional encoding methods, introduces a relative temporal encoding technique to handle dynamic graphs. Wang et al. (2022a) addressed the task of few-shot link prediction over temporal KGs using a meta-learning-based approach that builds representations of new nodes by aggregating features of existing nodes within a specific $\Delta_t$ temporal neighborhood. Though these methods have empirically shown their prediction power, they struggle to easily incorporate state-of-the-art designs from static GNNs (e.g. skip connections), which are beneficial for GNN architectural design (You et al., 2022; Xu et al., 2021). Furthermore, most of the works use only a fixed-split setting (You et al., 2022) to evaluate link

prediction performance or do not evaluate it at all. A fixed-split setting does not take into account the evolving nature of data as it provides to train the model on a huge part of historical information and test it only on the last timestamped information. In contrast, the recently proposed live-update setting (You et al., 2022), where models are trained and tested over time, can lead to a better evaluation for temporal graph learning models since performances are measured for each test snapshot.

**Factorization-based models.** Factorization-based Models (FMs) have enjoyed enduring success in Knowledge Graph Completion (KGC) tasks, often outperforming GNNs (Chen et al., 2022). Various FMs have been proposed for temporal KGs (Cai et al., 2022). Despite their huge prediction power reached with simple architecture and order of magnitude fewer parameters compared to GNNs, they have shown a few drawbacks; for instance, they struggle to incorporate node features, they work in transductive settings only, and they are heavily designed to cope only with KGC tasks.

DURENDAL differs from the above works proposing a new update scheme for node embeddings that preserve heterogeneous information from the past and capture relational temporal dynamics. Moreover, it can handle node features and inductive tasks w.r.t. FM models since it relies on GNN architectures. Lastly, DURENDAL can be trained and evaluated in a live-update setting You et al. (2022) that takes into account the evolving nature of the data.

## 3 THE PROPOSED FRAMEWORK: DURENDAL

**Temporal heterogeneous graphs.** A heterogeneous graph, denoted as $G = (V, E)$, consists of a set of nodes $V$ and a set of links $E$. A heterogeneous graph is also associated with a node-type $\phi : V \mapsto A$ and a link-type $\psi : E \mapsto R$ mapping functions, where $A$ and $R$ are the predefined sets of node and link types such that $|A| + |R| > 2$. Nodes can be paired with features related to a certain node type $X_a = \{x_v \mid v \in V \wedge \phi(v) = a\}$. On the other hand, in a temporal graph, each node $v$ has a timestamp $\tau_v$ and each edge $e$ has a timestamp $\tau_e$. We focus on the snapshot-based representation of temporal graphs, at the basis of the definition of temporal heterogeneous graph. In fact, a temporal heterogeneous graph $\mathcal{G} = \{G_t\}_{t=1}^{T}$ can be represented as a sequence of graph snapshots, where each snapshot is a heterogeneous graph $G_t = (V_t, E_t)$ with $V_t = \{v \in V | \tau_v = t\}$ and $E_t = \{e \in E | \tau_e = t\}$.

**Heterogeneous GNNs.** The objective of a GNN is to learn node representations via an iterative aggregate of neighborhood messages. In heterogeneous graph learning, models exploit the highly multi-relational data characteristic as well as the difference in the features related to each node type, to obtain better representations of nodes. Hence, in heterogenous GNNs node embeddings are learned for each node type and messages are exchanged between each edge type. Then, the partial node representations derived for each edge type, in which they are involved, are mixed together through an aggregation scheme. Formally, we denote by $H^{(L)} = \{h_v^{(L)}\}_{v \in V}$ the embedding matrix for all the nodes after applying an $L$-layer GNN. The $l$-layer of a heterogenous GNN, $H^{(l)}$, can be written as:

$$h_v^{(l)} = \bigoplus_{r \in R} f_\theta^{(l,r)}(h_v^{(l-1)}, \{h_w^{(l-1)} : w \in \mathcal{N}^{(r)}(v)\})$$

where $\mathcal{N}^{(r)}(v)$ denotes the neighborhood of $v \in V$ under relation $r \in R$, $f_\theta^{(l,r)}$ denotes the message passing operator for layer $l$ and relation $r$, and $\bigoplus$ is the aggregation scheme to use for grouping node embeddings generated by different relations. In the following sections, we will also refer to partial views of the embedding matrix w.r.t. types. Specifically, we will use $H^{(l,r)}$ to denote the partial embeddings related to a relation type $r \in R$ and $H^{(l,a)}$ to denote the node embedding matrix related only to a specific node type $a \in A$.

**From heterogeneous GNNs to temporal heterogeneous GNNs.** Figure 1 shows the proposed DURENDAL framework to generalize any heterogenous GNNs to a dynamic setting. Following the ROLAND (You et al., 2022) model design principle, the node embeddings at different GNN layers are hierarchical node states which are recurrently updated over time through customizable embedding modules. To allow easy repurposing of any heterogenous GNNs to a temporal setting, we introduce heterogeneous hierarchical node states and customizable semantic aggregation schemes, that define how partial node representations for each relation type are aggregated. In this way, modern architectural design options such as skip connections or attention mechanisms can be easily incorporated. Node embeddings can be updated using a moving average, a two-layer MLP, or a GRU

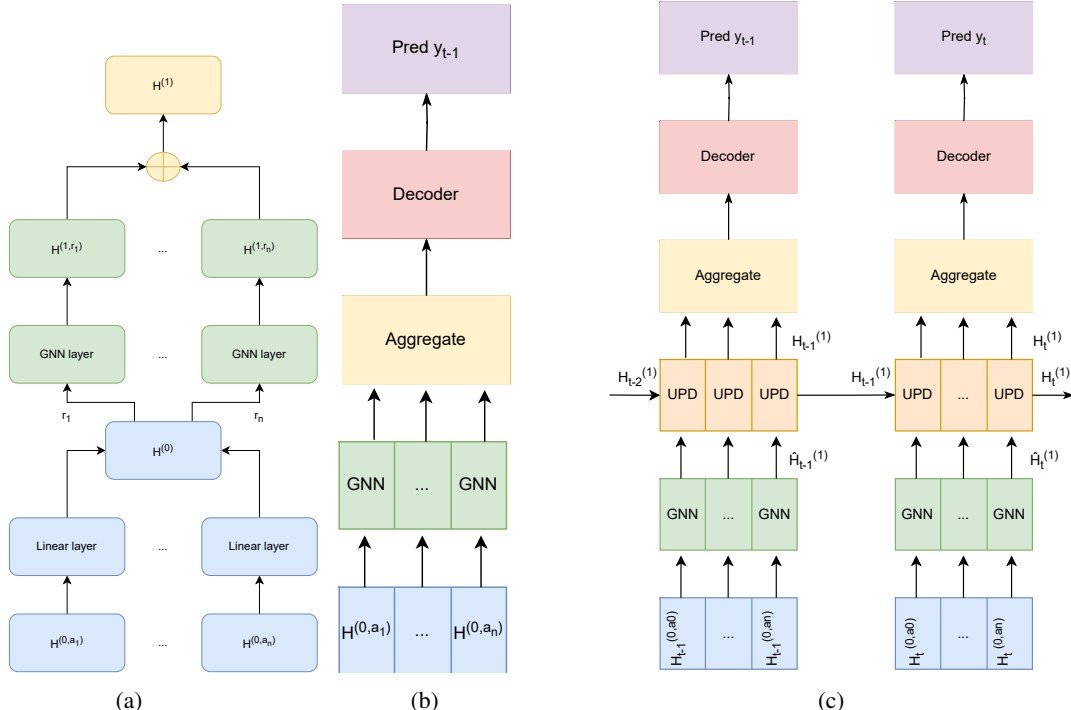

Figure 1: DURENDAL model design. (a) Scheme of the computation beyond a heterogeneous GNN layer. (b) Compact representation of the (a) scheme within the GRAPHEDM paradigm (Chami et al., 2022). (c) DURENDAL framework with the *Update-Then-Aggregate* scheme: the orange layer (temporal layer) updates over time the hierarchical node state of each relation type (returned by the first two layers in (b)), then the aggregation scheme (yellow) is run on top the temporal layer. In the *Aggregate-Then-Update* scheme the temporal layer and the aggregation scheme are swapped.

Cell. A suitable option for the semantic aggregation scheme could involve semantic-level attention coefficients (Wang et al., 2019). The forward computation of the $l-$layer of DURENDAL on the $t$ snapshot for a node $v$, $h_{v_t}^{(l)}$, can be written as:

$$h_{v_t}^{(l)} = \bigoplus_{r \in R} \text{UPDATE}(f_\theta^{(l,r)}(h_{v_t}^{(l-1)}, \{h_w^{(l-1)} : w \in \mathcal{N}^{(r,t)}(v)\}), h_{v_{t-1}}^{(l)}) \tag{1}$$

where UPDATE (UPD in Figure 1) is a custom update function and $\mathcal{N}^{(r,t)}(v)$ is the neighbourhood of $v$ on the relation $r$ at time $t$.

**Updating schemas: Update-Then-Aggregate and Aggregate-Then-Update.** As shown in Eq. 1, node states are first updated over time and then aggregated along the different semantic levels, i.e. relation types. We denote this solution as *Update-Then-Aggregate* scheme - UTA. This scheme provides a rich representation of temporal heterogeneous information. Indeed, it captures relational temporal dynamics by preserving partial node states that are updated through several embedding modules, one for each relation type. Furthermore, thanks to the heterogeneous node states, it is more suited for continual learning (Yuan et al., 2023) settings and it allows partial update scenarios, i.e. feeding the model with a new batch of data related to a specific subset of relations or node types. In contrast, an *Aggregate-Then-Update* (ATU) scheme can be used to first aggregate the partial representation of nodes and then update the node states using a single update module. Formally, the forward computation of DURENDAL with the *Aggregate-Then-Update* scheme can be written as:

$$h_{v_t}^{(l)} = \text{UPDATE}(\bigoplus_{r \in R} f_\theta^{(l,r)}(h_{v_t}^{(l-1)}, \{h_w^{(l-1)} : w \in \mathcal{N}^{(r,t)}(v)\}), h_{v_{t-1}}^{(l)}) \tag{2}$$

This second solution loses the heterogeneity of the information from the past because it updates the node embeddings only at the semantic-aggregated level. However, it is useful to reduce the memory

footprint of the model when modeling relational temporal dynamics is not beneficial (see Appendix for use case examples). Moreover, utilizing a single embedding update module reduces the number of learnable parameters, thereby mitigating the model's susceptibility to overfitting.

**Scalability.** To train DURENDAL, it is not necessary to keep in memory the whole temporal heterogeneous network. Indeed, we use the live-update setting for training DURENDAL. The live-update setting is an incremental training approach in which the model is fine-tuned and then tested on each snapshot. Hence, given a new graph snapshot $G_t$, since the hierarchical node states $H_{t-1}$ have already encoded information up to time $t-1$, to train the model and make predictions on $t$ only $G_t$, the GNN model and $H_{t-1}$ must be stored in the C/GPU memory. In addition, if we adopt the *Update-Then-Aggregate* scheme, we can easily split the computation for each relation type from the input until the aggregation layer. This splitting allows us to *i)* parallelize the training procedure on the different semantic levels of the network; and *ii)* keep in memory only a portion of the GNN model, node states, and new data related to a specific semantic level.

## 4 TEMPORAL HETEROGENEOUS NETWORKS DATASET

Here we present four THN datasets to evaluate the performance of graph machine-learning models on future link prediction tasks. The datasets serve as useful playgrounds for testing graph ML models because they provide high-resolution temporal heterogeneous information along multiple time snapshots. To the best of our knowledge, there are no current benchmark datasets for temporal heterogeneous graph learning.

**Dataset requirements.** We define some minimal requirements graph datasets must meet to be considered suitable for temporal heterogeneous graph learning evaluation. Specifically, we introduce three simple metrics to measure different properties of the data: *heterogeneity*, *temporality*, and *evolutivity*. Heterogeneity is the number of relation types available in the dataset, temporality is the number of graph snapshots, and evolutivity is the average number of new links in the snapshots (i.e. $\frac{1}{|T-1|} \sum_{t=1}^{T} |E_t|$). We require a value for heterogeneity greater or equal (g.e.q.) to two (by definition of heterogeneous graphs), for temporality g.e.q. to four (minimum number of snapshots to allow live-update evaluation (You et al., 2022)), and for evolutivity g.e.q. to zero (i.e. edges have timestamps). Furthermore, we define *time-granularity* as the duration of the time interval on which a graph snapshot is constructed, but we do not impose a minimum value for this metric.

**Our datasets.** To cope with the above issue, we present four THN datasets that satisfy our requirements. The first two datasets are part of a well-established suite for benchmarking knowledge base completion tasks, while the remaining two are introduced in this work to extend the benchmark set for THNs.

- `GDELT18, ICEWS18`: the Global Database of Events, Language, and Tone and the Integrated Crisis Early Warning System used by Jin et al. (2020). The two datasets are event knowledge graphs in which nodes are actors and edges are verbs between actors. They are used to evaluate temporal knowledge base completion (Cai et al., 2022) tasks and are available in the most used graph representation learning libraries. We process the data according to Jin et al. (2020) and then we construct graph snapshots with time granularity equal to one week for GDELT and to one month for ICEWS. Since most of the verbs have no instances in the original datasets, we decided to select only the top 20 most frequent verbs. Actors and verbs codename follow the CAMEO ontology.[1]

- `TaobaoTH`: a transformation of a dataset of user behaviors from Taobao, offered by Alibaba and provided by the Tianchi Alicloud platform[2]. The original dataset was used in prior works on recommendation (Zhu et al., 2019) or temporal graph learning (Jin et al., 2022) but user behaviors were considered just as node features or not considered at all. Here we transform Taobao data into a temporal heterogeneous graph for recommendation, where the types of nodes are users, items, and categories for items. To introduce heterogeneity, edges between users and items represent different types of user actions (buy, pageview, add to favorite/cart) towards items - with timestamps. Moreover, edges between items and categories assign

---

[1] `https://parusanalytics.com/eventdata/data.dir/cameo.html`, September 2023
[2] `https://tianchi.aliyun.com/dataset/649`, September 2023

each item to its set of categories. We construct heterogeneous graph snapshots with time granularity equal to five minutes. We consider a heterogeneous subgraph induced by 250k random sampled items for scalability issues.

- `SteemitTH`: A novel benchmark dataset collecting different kinds of user interactions from Steemit (Li et al., 2021a), the most well-known blockchain-based online social network. Users on Steemit have access to a wide range of social and financial operations that keep track of user activity with a *three-second* temporal precision on the underlying blockchain. Data from June 3, 2016, through February 02, 2017, have been gathered through the official API which provides access to the information stored in the blocks. For building a heterogeneous graph, we focused on four kinds of relationships: "follow", upvote (like), comment, and Steem Dollars (SBD) transfer - financial operation. The heterogeneous graph snapshots have a monthly time granularity. The starting date corresponds to when the "follow" operation has been made available on the platform. We also collected the textual content produced by users, used to build a feature vector for each node (more details in the Appendix)

We report some dataset statistics in Table 1. The number of nodes and edges refers to the whole graph.

Table 1: Dataset statistics. Evolutivity is divided by $|E|$.

| Dataset | |N| | |E| | |R| | |T| | time-granularity | evolutivity |
|---|---|---|---|---|---|---|
| GDELT18 | 4,931 | 2,026 | 20 | 4 | week | 0.263 |
| ICEWS18 | 11,775 | 7,333 | 20 | 7 | month | 0.139 |
| TaobaoTH | 359,997 | 210,448 | 5 | 288 | 5min | 0.003 |
| SteemitTH | 20,849 | 1,832,570 | 4 | 5 | month | 0.177 |

## 5 EXPERIMENTAL EVALUATION

**Tasks.** The future link prediction problem arises in many different applications and domains. When it comes down to heterogeneous graphs, link prediction can be performed on the whole set of relation types (e.g. Knowledge Base Completion (Cai et al., 2022), multirelational link prediction (Zitnik et al., 2018)) or on a specific relation. We conducted our evaluation considering both kinds of link prediction tasks. Specifically, given all the graph snapshots up to time $t$ and a candidate pair of nodes $(u, v)$, the *monorelational future link prediction* task consists of finding if $(u, v)$ are connected through a given relation $r$ in a future snapshot $t + 1$; while the *multirelational future link prediction task* involves any $r$ in the set of all the possible relations. For the monorelational tasks, we focus on a specific relation type for each dataset to study how the models can learn from past information and current heterogeneous interactions between nodes to predict predefined future relations. This choice allows us to analyze the prediction performance in real-application scenarios on general heterogeneous graphs, i.e. graphs that are not KGs, as in the case of `SteemitTH` and `TaobaoTH`. Specifically, we perform the following future link prediction tasks: *i)* "follow" link prediction between users for `SteemitTH`; *ii)* "buy" link prediction between users and items for `TaobaoTH`; and *iii)* public statements prediction from one actor to another (e.g. governments) for `GDELT18` and `ICEWS18`, according to the CAMEO ontology. For the multirelational tasks, we focus on the event KGs as they represent two standard benchmark datasets for heterogeneous graph learning. Moreover, considering problems different from "user-follow-user" and "user-buy-item" prediction could be not so interesting and meaningful for `SteemitTH` and `TaobaoTH`.

**Experimental setup.** We evaluate the DURENDAL framework over the future link prediction task. At each time $t$, the model utilizes information up to time $t$ to predict edges in the snapshot $t + 1$. We use the area under the precision-recall curve (AUPRC) and the mean reciprocal rank (MRR) to evaluate the performance of the models. As a standard practice (Pareja et al., 2020), we perform random negative sampling to obtain an equal number of positive and negative edges[3]. We consider the live-update setting (You et al., 2022) for the evaluation of the models by which we assess their performance over all the available snapshots. We randomly choose 20% of edges in each snapshot to determine the early-stopping condition. It is worth noting that in `SteemitTH` we also use the node

---

[3]We sampled negative edges due to memory constraints.

features derived from the textual content, while in the other settings, node features are not available. We rely on HadamardMLPs (Wang et al., 2022b) and ComplEx (Trouillon et al., 2016) as decoders for monorelational and multirelational link prediction as both demonstrated their effectiveness compared to other link prediction decoders (Wang et al., 2022b; Ruffinelli et al., 2020; Lacroix et al., 2018). For the multirelation link prediction experiments, we rely on the experimental setting presented by Zitnik et al. (2018). We compute the AUPRC and MRR score for each relation type, averaging the performance over all the relations to obtain the final evaluation scores. To extend this setting to THNs, we repeat the procedure for each snapshot using the live-update evaluation. Code, datasets, and all the information about the experiments are available in our repository[4].

**Baselines.** We compare DURENDAL to nine baseline models, considering at least one candidate for homogeneous, heterogeneous, static, and dynamic graph learning. Among the static graph learning models, we decide to compare DURENDAL with solutions that utilize an attention mechanism, whose great potential has been well demonstrated in various applications (Hu et al., 2018; Lee et al., 2019; Hu et al., 2020b). Whereas for temporal graph learning models, we compare the performance with temporal GNNs (Longa et al., 2023) as well as walk-aggregating methods (Kazemi et al., 2020). Specifically, we select the following candidates: *GAT* (Veličković et al., 2018), *HAN* (Wang et al., 2019), *EvolveGCN* (Pareja et al., 2020), *GCRN-GRU*, *TGN* (Rossi et al., 2020), *CAW* (Wang et al., 2021), and HetEvolveGCN (a new baselines we developed for snapshot-based THNs, see Appendix). For multirelational link prediction, baselines need to leverage heterogeneous graphs. Hence, we consider HAN, HetEvolveGCN, and two additional baselines based on tensor factorization, which demonstrated huge prediction power on knowledge graphs link prediction (Ruffinelli et al., 2020; Lacroix et al., 2018; Cai et al., 2022): *ComplEx* (Trouillon et al., 2016) and *TNTComplEx* (Lacroix et al., 2020). A brief description of the baselines is provided in the Appendix. All the candidate models have been trained using the incremental training procedure.

**Results for monorelational link prediction.** Table 2 shows the prediction performance of the candidate models in monorelational future link prediction tasks. We report the average AUPRC and MRR over the different snapshots. DURENDAL achieves better performance compared to baselines in three of the four datasets. On `GDELT18` and `ICEWS18`, all dynamic models achieve performances around 90% because they leverage temporal information related to events, which is crucial for predicting future public statements. DURENDAL, achieving the best performance overall, gains the greatest advance from the semantics related to events, i.e. the different relation types. On `SteemitTH`, all the models obtain great performances; DURENDAL, by exploiting information derived from node attributes, timestamps on edges, and semantic relations, reaches an AUPRC and MRR score of $0.982$ and $0.891$, respectively. On `TaobaoTH`, we obtain surprising results. The best performance is achieved by HAN, that do not use leverage temporal information, apart from the incremental training. TGN and CAW achieve notably worse prediction performance than heterogeneous GNNs, while EvolveGCN, GCRN-GRU, and HetEvolveGCN obtain poor performance. DURENDAL reaches good performance using an embedding update module that simply computes a convex combination between the past and the current representation of nodes, with a past coefficient no greater than $0.1$. The same results are obtained using a time granularity of one or ten minutes. Hence, predicting future "buy" relations seems just related to the other actions performed by users on items (view an item, add it to your favorites or in your cart) in the previous snapshot, not to the order they are carried out, nor to repetition over time. The result is surprising because sophisticated dynamic models seem to give too much importance to past information without learning this simple structural pattern. However, it is important to note that `TaobaoTH` has a very low evolutivity value, equal to $0.003$. Finally, it is worth noticing that TGN and CAW reach worse performance of at least one snapshot-based baseline for three of the four datasets. In our intuition, their continuous-time representation for temporal networks is not beneficial in application scenarios where datasets are snapshots-based.

**Results for multirelational link prediction.** Table 3 shows the prediction performance of the candidate models in multirelational future link prediction tasks. We report the average AUPRC and MRR over the different snapshots. DURENDAL performs better than baselines on both datasets with at least one of the two update schemes. The results highlight the importance of two different update schemes for temporal knowledge graph forecasting (Gastinger et al., 2023). On `GDELT18`, the best performance is achieved using the *Upgrade-Then-Aggregate* scheme, i.e. preserving partial node

---

[4]https://anonymous.4open.science/r/durendal-5154/

Table 2: Results on the monorelational future link predictions tasks in terms of AUPRC and MRR averaged over time. We run the experiments using 3 random seeds, reporting the average result for each model. Results for TGN and CAW are obtained using their official implementations in a live-update setting[5].

|  | GDELT18 | | ICEWS18 | | TaobaoTH | | SteemitTH | |
| --- | --- | --- | --- | --- | --- | --- | --- | --- |
|  | AUPRC | MRR | AUPRC | MRR | AUPRC | MRR | AUPRC | MRR |
| GAT | 0.488 | 0.506 | 0.477 | 0.506 | 0.500 | 0.500 | 0.940 | 0.845 |
| HAN | 0.564 | 0.601 | 0.561 | 0.566 | **0.996** | **0.996** | 0.974 | 0.859 |
| EvolveGCN | 0.933 | 0.864 | 0.930 | 0.898 | 0.500 | 0.500 | 0.979 | **0.895** |
| GCRN-GRU | 0.935 | 0.806 | 0.873 | 0.816 | 0.500 | 0.500 | 0.950 | 0.855 |
| TGN | 0.908 | - | 0.916 | - | 0.710 | - | 0.889 | - |
| CAW | N/A | - | 0.893 | - | 0.518 | - | 0.907 | - |
| HetEvolveGCN | 0.877 | 0.855 | 0.934 | 0.922 | 0.5 | 0.5 | 0.977 | 0.879 |
| DURENDAL | **0.947** | **0.930** | **0.986** | **0.981** | 0.995 | 0.993 | **0.982** | 0.891 |

Table 3: Results on the multirelational future link predictions tasks in terms of AUPRC and MRR averaged over time. We report the average result for each method over experiments with 3 different random seeds.

|  | GDELT18 | | ICEWS18 | |
| --- | --- | --- | --- | --- |
|  | AUPRC | MRR | AUPRC | MRR |
| HAN | 0.608 | 0.704 | 0.618 | 0.710 |
| HetEvolveGCN | 0.628 | 0.664 | 0.611 | 0.653 |
| ComplEx | 0.527 | 0.705 | 0.505 | 0.699 |
| TNTComplEx | 0.540 | **0.744** | 0.525 | 0.743 |
| DURENDAL-UTA | **0.672** | 0.743 | 0.677 | 0.745 |
| DURENDAL-ATU | 0.660 | 0.730 | **0.693** | **0.749** |

states to capture relational temporal dynamics. Indeed, due to the significant temporal variability of the Global Database of Events, datasets extracted from GDELT are considered more challenging than the ones collected from ICEWS (Messner et al., 2022; Wu et al., 2020). GDELT18 exhibits also the highest evolutivity rate in Table 1. Hence, using different embedding update modules for different relations is beneficial to predict its evolution. On ICEWS18, preserving partial node states leads to slightly worse results. In our intuition, as highlighted for other datasets collected from the ICEWS system, ICEWS18 requires more entity-driven predictions, as the relations in these datasets are sparse and they usually encode one-time patterns with limited, if any, regularity, e.g., official visits, or negotiations (Messner et al., 2022). Hence, by focusing on the evolution of the entity embeddings instead of modeling relational temporal dynamics, DURENDAL with the *Aggregate-Then-Update* scheme achieves the best results. It is worth noting that factorization-based models, typically used for temporal knowledge graph completion (Cai et al., 2022) (i.e. missing temporal link prediction), achieve good performance on these temporal knowledge graph forecasting tasks, often outperforming other GNN baselines on both datasets.

**Effectiveness of model-design.** DURENDAL can easily repurpose any heterogenous GNN to a dynamic setting thanks to its model design. Here we study the prediction performance of different DURENDAL repurposed heterogeneous GNNs. Specifically, we repurpose RGCN (Schlichtkrull et al., 2018), HAN (Wang et al., 2019), and HGT (Hu et al., 2020b). Node embeddings are updated using ConcatMLP You et al. (2022) or a GRU cell, following the *Aggregate-Then-Update* scheme. Figure 2a shows the AUPRC distributions of the models on the "follow" link prediction task on SteemitTH. Results show that an attention-based aggregation scheme for heterogeneous graph learning is a valuable choice for GNN architectural design. Indeed, HAN and HGT achieve the best results and their AUPRC distributions exhibit low variance. Furthermore, ConcatMLP seems preferable to a GRU Cell because it obtains better results with negligible variation. Lastly, the

---

[5]The official implementations for TGN and CAW do not compute the MRR for evaluating their performance. On GDELT18, CAW obtains nan values as AUPRC score.

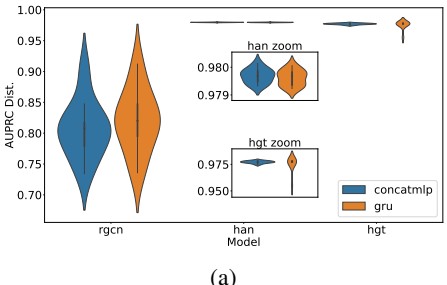
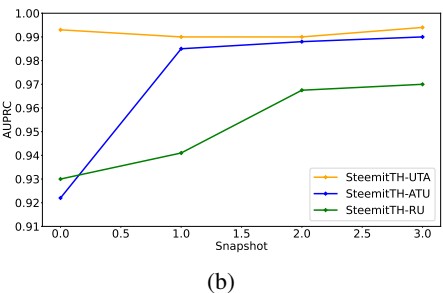

(a)                                                                          (b)

Figure 2: (a) AUPRC distributions of DURENDAL repurposed RGCN, HAN, and HGT on future link prediction task on `SteemitTH`, using ConcatMLP or a GRU cell as embedding update module (experiments with 10 random seeds). Attention-based aggregation schemes and ConcatMLP update modules are desirable for GNN architectural design. (b) Results snapshot-by-snapshots of DUREN-DAL models for "follow" link prediction on `SteemitTH` using different update schemes. UTA outperforms the other update schemes but ATU is still a profitable choice for learning on multiple snapshots.

Table 4: Results comparison between UTA, ATU and ROLAND-update (RU) schemes. We report the average result for each method over experiments with 3 different random seeds .

|       | GDELT18 |       | ICEWS18 |       | TaobaoTH |       | SteemitTH |       |
|-------|---------|-------|---------|-------|----------|-------|-----------|-------|
|       | AUPRC   | MRR   | AUPRC   | MRR   | AUPRC    | MRR   | AUPRC     | MRR   |
| UTA   | 0.947   | 0.930 | 0.986   | **0.981** | 0.995    | 0.993 | **0.982** | **0.891** |
| ATU   | **0.996** | **0.962** | **0.991** | 0.978 | **0.999** | **0.999** | 0.971     | 0.882 |
| RU    | 0.935   | 0.806 | 0.875   | 0.816 | 0.500    | 0.500 | 0.950     | 0.855 |

DURENDAL model design helps HAN to reach better results: the worst result in its AUPRC distribution is $0.979$, which is better than the average result of "vanilla" HAN $0.974$ (see Table 2).

**Effectiveness of update schemes.** We also studied the effectiveness of the two different update schemes described in Section 3. Table 4 reports the prediction performance of DURENDAL models with *Update-Then-Aggregate*, *Aggregate-Then-Update*, and ROLAND-update, i.e. no heterogeneous update. The update schemes of DURENDAL perform better than the ROLAND update scheme. In particular, *Update-Then-Aggregate* seems preferable to *Aggregate-Then-Update* when the time granularity of the dataset is coarser, and vice-versa. Finally, we also show the prediction performance snapshot by snapshots for "follow" link prediction on `SteemitTH` in Figure 2b. In this context, *UpdateThen-Aggregate* dominates the other update schemes but *Aggregate-Then-Update* is still a profitable choice for learning on multiple snapshots.

## 6 CONCLUSION

We propose DURENDAL, a snapshot-based graph deep learning framework for learning from temporal heterogeneous networks. Inspired by the ROLAND framework for dynamic homogeneous graphs, DURENDAL can help to easily repurpose any heterogeneous graph learning model to dynamic graphs, including training strategies and evaluation settings for evolving data. To help easy repurposing, DURENDAL introduces heterogeneous hierarchical node states and customizable semantic aggregation schemes. We also introduce two different update schemes, highlighting the strengths and weaknesses of both in terms of scalability, memory footprint, and learning power. To evaluate our framework, we describe the minimum requirements a benchmark should satisfy to be a useful testing ground for temporal heterogenous GNN models, and we extend the current set of benchmarks for TNHs by introducing two novel high-resolution temporal heterogeneous graph datasets. We evaluate DURENDAL over the future link prediction task using incremental training and live-update evaluation over time. Experiments show the prediction power of DURENDAL over four THNs datasets, which exhibit different time granularity, number of snapshots, and new incoming links. Moreover, we show the effectiveness of the DURENDAL model design by enhancing the prediction performance of heterogenous GNN models by repurposing them in our framework.

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

## A APPENDIX

**Description of the baselines.** Here we briefly summarize the baseline methods we used in the evaluation and comparison.

1. *GAT* (Veličković et al., 2018) for homogeneous graphs. It leverages masked self-attentional layers with graph convolution to implicitly obtain different weights for different nodes in a neighborhood;

2. *HAN* (Wang et al., 2019) for heterogeneous graphs. It introduces hierarchical attention, including node-level and semantic-level attention. Specifically, node-level attention aims to learn the importance between a node and its metapath-based neighbors, while semantic-level attention is able to learn the importance of different meta-paths.

3. *EvolveGCN* (Pareja et al., 2020) for dynamic graphs. It utilizes an RNN to dynamically update the weights of internal GNNs, which allows the GNN model to change during test time.

4. *GCRN-GRU* (Seo et al., 2018) for dynamic graphs. It is a generalization of the T-GCN model Zhao et al. (2020), which internalizes a GNN into the GRU cell by replacing linear transformations in GRU with graph convolution operators. GCRN uses ChebNet Defferrard et al. (2016) for spatial information and separate GNNs to compute different gates of RNNs.

5. *TGN* (Rossi et al., 2020) for dynamic graphs. Temporal Graph Networks (TGNs) adopt a continuous-time representation for temporal networks (Longa et al., 2023) by treating the interactions between nodes as a list of events that occur over time. When two nodes are involved in an interaction, they exchange messages (Gilmer et al., 2017), which are then used to update their memories, vectors that serve as a condensed representation of a node's historical interactions at a specific point in time. To obtain the final node's embedding, an extra graph aggregation step is conducted involving the node's temporal neighbors, incorporating both the node's original features and its memory state at a specific point in time.

6. CAW (Wang et al., 2021) for dynamic graphs. Causal Anonymous Walks (CAW) leverage temporal random walks to perform link prediction adopting a continuous time representation for temporal networks (Longa et al., 2023). Specifically, given a candidate pair $(u, v)$ at time $t$, CAW extracts multiple random walks starting from $u$ and $v$ such that the timestamps of edges in a walk can only be monotonically decreasing. Subsequently, the walks undergo anonymization, where every node identifier is substituted with a count vector indicating how frequently that node appears at various positions within the walks. Each walk is then encoded using an RNN, and the resulting encodings are aggregated either through self-attention or by computing a simple average.

7. *HetEvolveGCN* for THNs. It is an extension of the EvolveGCN model we developed to deal with temporal heterogeneous graphs in a live-update setting. It applies EvolveGCN

for each metapath, i.e. it has an RNN to dynamical update weights of each relation-type specific GNN. The partial representations are then aggregated using summation.

8. ComplEx (Trouillon et al., 2016) for static heterogeneous graphs. ComplEx introduces complex embeddings for entities and relations in knowledge graphs and the Hermitian dot product as scoring functions to perform link prediction using tensor factorization.

9. TNTComplEx (Lacroix et al., 2020) for temporal heterogeneous graphs. TNTComplEx extends the ComplEx decomposition to the temporal link prediction setting by adding a new factor $T$ related to timestamp embeddings. To allow factorization-based models to work with unseen timestamps (i.e. the temporal knowledge graph forecasting setting (Gastinger et al., 2023)), we apply a recurrent architecture to model the temporal dynamics as in Dileo et al. (2023).

**Live-update setting.** As the live-update setting is a newly introduced protocol for training and evaluating temporal graph learning models, we briefly summarize its steps as follows:

1. Split the dataset of the current snapshot into train and validation set.

2. Train the model on the train set optimizing the binary cross-entropy loss until the prediction performance on the validation set does not get worse (early-stopping condition).

3. Compute the prediction performance on the next snapshot.

4. Repeat steps 1-3 from the first snapshot until the second-last fine-tuning the model on the new snapshot.

5. Compute the overall prediction performance of the model over time by averaging the performance over the single snapshots, or analyze the performance snapshot-by-snapshot.

**Use case examples for UTA and ATU.** In our work, we propose two different schemes for the node embedding updates over time. The first, namely *Update-Then-Aggregate* (UTA), provides a rich representation of temporal heterogeneous information as it preserves partial node states that are updated through several embedding modules, one for each relation type. The latter, *Aggregate-Then-Update* (ATU), uses a single embedding update module as it updates the node embedding at the semantic-aggregated level. In the following, we provide some examples of when UTA could be a more profitable choice than ATU and vice-versa.

- UTA is useful for capturing relational temporal dynamics in application scenarios where relations between entities evolve with different characteristics and temporal scales. On `SteemitTH`, we observed that UTA performs better than ATU as this dataset, derived from a blockchain-based online social network, exhibits both social - "follow" relationships - and economic interactions - cryptocurrency transactions- which shows different temporal patterns and an unstable correlation over time. Indeed, economic transactions between users occur frequently, with different amounts of exchange and repeatedly, whereas "follow" relationships typically occur one-time between two users and do not have implicit weights, i.e. all the relationships a user creates have the same "weight". Another interesting use case for UTA could be a temporal heterogeneous network related to the IMDB dataset [6]. In fact, a relationship between Actor and Movie entities, such as `Actor-starred_in-Movie`, occurs densely and with variability over time, while a relationship between actors, such as `Actor-married-Actor`, occurs rarely and at most only a few time for each entity.

- ATU could be a profitable choice when relations do not exhibit very different temporal patterns and scales. For instance, `ICEWS18` contains mainly relations that are sparse and they usually encode one-time patterns with limited, if any, regularity, e.g., official visits, or negotiations, and ATU performs better than UTA. As a further example, a subset of `SteemitTH` with vote and comment relationships only may also benefit from ATU as these two social interactions are very similar.

**Training details for GNN-based architectures.** We developed DURENDAL using Pytorch Geometric (PyG) (Fey & Lenssen, 2019). We use the implementation available in PyG for GAT, HAN, RGCN, and HGT. For EvolveGCN, GCRN-GRU, and HetEvolveGCN, we use the implementation available

---

[6]`https://developer.imdb.com/non-commercial-datasets/`, September 2023

in Pytorch Geometric Temporal (Rozemberczki et al., 2021). We used the official implementations for TGN and CAW. We ran our experiments on NVIDIA Corporation GP107GL [Quadro P400]. In all our experiments, we use the Adam (Kingma & Ba, 2015) optimizer. This choice was made according to some prior works on GNN architecture for temporal heterogeneous networks (Jin et al., 2020; Li et al., 2021b; Wang et al., 2019; Hu et al., 2020b). We adopt the live-update setting to train and evaluate the models, wherein we engage in incremental training and assess their performance across all the available snapshots. With respect to each snapshot, a random selection of 20% of edges is employed to establish the early-stopping condition (validation set), while the remaining 80% are utilized as the training set. The edges contained within the subsequent snapshot constitute the test set. Consistently, we apply identical dataset divisions and training procedures across all the methods. Hyperparameters are tuned by optimizing the AUPRC on the validation set, and the model parameters are randomly initialized. The hyperparameter search spaces are as follows: learning rate {0.1, 0.01, 0.001}, L2 weight-decay {5e-1, 5e-2, 5e-3}, number of hidden layers {1, 2}, representation dimension {32, 64, 128, 256}. For DURENDAL, we also tested three different message-passing operators: GAT (Veličković et al., 2018), SAGE (Hamilton et al., 2017), and the operator from the work by Morris et al. (2019).

**Implementation details for factorization-based models.** Factorization-based models (FMs) achieve state-of-the-art performances on several benchmark datasets for both static and temporal knowledge graph completion (Lacroix et al., 2018; Cai et al., 2022). Despite their huge prediction power, current implementations are specifically designed to solve completion tasks, i.e. predicting missing links, and they cannot perform predictions on unseen timestamps. Moreover, to the best of our knowledge, the prediction power of FMs on temporal knowledge graph forecasting tasks (Gastinger et al., 2023), like those presented in this work, has not been analyzed. To cope with these problems, we implemented a new version of TNTComplEx where timestamp embeddings can be generated sequentially using a recurrent neural architecture. The idea was inspired by Dileo et al. (2023) where an RNN is used for temporal regularization (Lacroix et al., 2020). Given the equation that describes TNTComplEx in (Lacroix et al., 2020), the tensor $T$ related to timestamps is generated row-by-row sequentially using a customizable recurrent architecture. The embedding $t_l$ of timestamp $l$ is obtained as:

$$\mathbf{t_l} = MLP(RNN(\mathbf{h_{l-1}}, \underline{\mathbf{0}})); l \in \{\mathbf{1}, ..., |\mathcal{T}|\} \tag{3}$$

where $h_0 \in \mathbb{R}^m$ is the learnable initial hidden state, $\underline{\mathbf{0}}$ is the zero vector, *RNN* is the function that describes the recurrent architecture, *MLP* is a function that describes one multi-layer perceptron layer that maps the output of the RNN to an output vector with the same embedding size of the entity and relation embeddings.

**Training details for FMs.** We used the implementation available in PyG for ComplEx and we implemented TNTComplEx on its top. We used Adagrad as the optimizer because it achieved state-of-the-art performances on several KGs datasets (Lacroix et al., 2018; 2020). The hyperparameter search spaces are as follows: learning rate {0.1, 0.01, 0.001}, L2 weight-decay {5e-1, 5e-2, 5e-3}, representation dimension {5, 25, 50, 100, 500, 2000}. For TNTComplEx, we tested RNN, LSTM, and GRU as recurrent architectures with hidden dimensions {5, 25, 50, 100, 500}. The best results reported in the paper are obtained using Adagrad with a learning rate equal to 0.1 and a weight-decay equal to 5e-3, GRU with 500 as hidden size, and an embedding dimension of 2000.

**GNN architectures.** We tested DURENDAL and all the other baselines using either one or two graph message-passing hidden layers. We do not test architecture with more than two graph message-passing layers to avoid the over-smoothing problem (Chen et al., 2020). We define and test two different configurations for the number of hidden neurons: the first has 64 and 32 hidden neurons and the second has 256 and 128. DURENDAL, GAT, and HAN achieve their best performances using 2 layers, while the other models use only one hidden layer. All the models reach better performance with the highest number of hidden neurons among the considered dimensions for each layer. For DURENDAL, we report in Table 5 the best configuration for the update modules for each dataset. In our intuition, ConcatMLP works better when the assumption of temporal smoothness, i.e. entities behave very similarly on neighboring timestamps, is strong. For example, this is the case of an online social network (Steemit). Vice versa, on event networks (ICEWS, GDELT), where entities are mainly government organization that acts in complex ways, a recurrent architecture that captures long-term patterns is beneficial. The weighted average, instead, can be leveraged when the actions performed by entities strongly depend on the previous snapshot, as highlighted in Section 5. The results presented in

the paper are obtained using a DURENDAL model with GraphConv as the message-passing operator and semantic-level attention mechanism (Wang et al., 2019) to aggregate partial node representations.

Table 5: Best embedding update module of DURENDAL models for future link prediction over the four THNs datasets.

| Dataset | Update module |
|---------|---------------|
| GDELT18 | GRU |
| ICEWS18 | GRU |
| TaobaoTH | Weighted average |
| SteemitTH | ConcatMLP |

**Resources and computational cost.** Table 6 reports the hardware specifics of the machine on which we run algorithms for data gathering, preprocessing, and all the experiments described in the paper. Overall, computing all the experiments with all the baselines and a single configuration of hyperparameters takes about 2 days. We describe the amount of computational time for individual experiments in Table 7. We report for each dataset the overall computational time of the pipeline from the data loading until the output of the prediction for each candidate model, considering one configuration of hyperparameters. A crucial part of the work is related to the data gathering and preprocessing but we do not report their computational time since we provide the obtained datasets. All the reported computational times are provided approximately.

| Resource | Description |
|----------|-------------|
| CPU | Intel Core i9-9820X CPU @ 3.30GHz x 20 |
| GPU | NVIDIA Corporation GP107GL [Quadro P400] |
| RAM | 64GB |
| Disk | 256 GB |

Table 6: The hardware specifications of the machine utilized for executing algorithms involved in data gathering, preprocessing, and all the experiments detailed in the paper.

| Experiment | Running time |
|------------|--------------|
| GDELT18 monorelational | 10min |
| ICEWS18 monorelational | 10min |
| SteemitTH | 30min |
| TaobaoTH | 20h |
| GDELT18 multirelational | 30min |
| ICEWS18 multirelational | 30min |
| Effectiveness update-scheme | 8h |
| Effectiveness model-design | 6h |

Table 7: Approximate computational time for individual experiments (overall computational time of the pipeline from the data loading until the output of the prediction for each candidate model).

Besides the approximate computational time for individual experiments, we also provide a discussion on the general computational efficiency of our solution. As DURENDAL is a general framework, the computation efficiency of our solutions depends on the efficiency of the underlying static heterogenous GNNs. In our experiments, we consider GraphConv (Morris et al., 2019) as message-passing operator and semantic attention (Wang et al., 2019) as aggregation mechanism. The number of parameters of GraphConv is linear in the number of edges, hence the total number of parameters for each heterogenous GNN layer is $|R| * \mathcal{O}(|\mathcal{E}|) + |R|$, where $R$ is the set of all possible relation and $E$ is the set of edges, the first addend is related to the message passing and the second to the single-head attention mechanism, which is again linear in the number of edges. The overhead given by DURENDAL is represented by the embedding updates modules, which contribute with a number of parameters equal to $|R|$ times the number of parameters of a single module (e.g. a GRU). Hence,

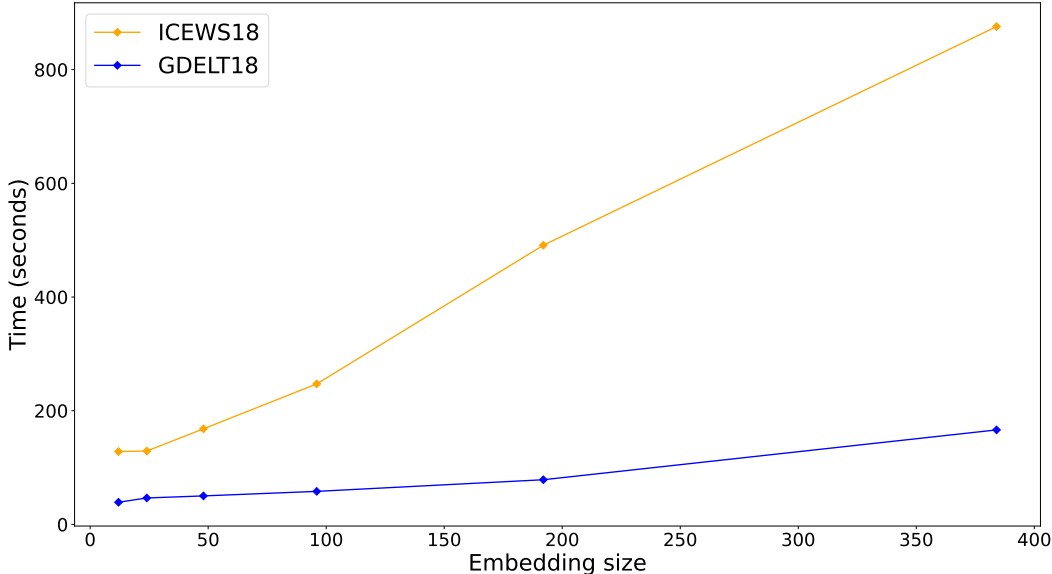

Figure 3: Execution time of DURENDAL on ICEWS18 and GDELT18 with different embedding size.

the computational efficiency of DURENDAL is strictly related to $|R|$. As described in Section 3, one can mitigate the computational cost by using an *Aggregate-Then-Update* scheme, which only requires a single embedding update instead of modules. It is also important to highlight that it is not necessary to keep in memory the whole temporal heterogeneous network and parameters (see scalability in Section 3). We also provide the training time of DURENDAL on different embedding sizes on ICEWS and GDELT, the two datasets with the biggest value for $|R|$, in Figure 3. Time grows no more than linear of the embedding size and our solution takes no more than 15 minutes for training.

**Dataset requirements: the case of OGB and TGB.** Open Graph Benchmark (OGB) (Hu et al., 2020a) is the most well-known benchmark dataset for graph ML. Despite its rich source of graph datasets, it does not contain networks that satisfy the minimum requirements we defined in the previous paragraph. Indeed, the only heterogeneous graphs available in the benchmark are `ogbl-biokg`, `ogbn-mag`, `ogbl-wikikg2`, and `WikiKG90Mv2`, but the first two datasets have evolutivity equal to zero, while the latter two have temporality equal to three. Recently, a temporal graph benchmark was publicly released (Huang et al., 2023). It contains several datasets for different temporal graph learning tasks, but none of them is heterogeneous.

**Task independence.** Following the encoder-decoder model Chami et al. (2022), DURENDAL can be used to learn node representations that are task-independent, useful to solve multiple learning tasks on graphs, or suited for specific tasks (e.g. dynamic node classification, link prediction, or graph classification). For instance, DURENDAL can be adopted to solve a node classification task by using a softmax function as decoder on the output layer and the cross entropy loss function. However, we have not tested DURENDAL on other learning tasks due to the absence of benchmark datasets. To the best of our knowledge, there is no dataset for dynamic node prediction or graph classification on a heterogeneous network. Moreover, as described above, in July 2023, a temporal graph benchmark was publicly released (Huang et al., 2023). It contains two datasets for dynamic node property prediction, but none of them is heterogeneous.

**Limitations.** In the evaluation setup, due to the high computation cost of evaluating all the possible candidates for each prediction, we adopted a random negative sampling strategy to compute MRR. That may lead to a biased evaluation metric since it is more likely to sample "easy" candidates due to the sparsity of real-world networks. Moreover, only in the last years, the research community is pushing towards better evaluation for temporal graph learning models Poursafaei et al. (2022). In the case of THNs these aspects require further investigation since current solutions have been

proposed only for homogeneous networks. Furthermore, current approaches are not suitable for the"incremental" temporal networks in our benchmark, such as online social networks. For instance, historical negative edges are not suitable since past "follow" links cannot be considered as a negative edge.

**Relations with other Graph Representation Learning techniques.** In this work we proposed a framework for temporal heterogeneous graph learning that is based on Graph Neural Networks (GNNs) as they represent the state-of-the-art for various graph machine learning tasks (Wu et al., 2021; Longa et al., 2023), they can leverage node features and deal with inductive tasks. When it comes down to heterogenous networks and knowledge graphs, there exist no GNN-based methods that work well on several benchmark datasets such as Metapath2Vec (Dong et al., 2017), which is based on random walks, and factorization-based models (Ruffinelli et al., 2020), such as DistMult (Yang et al., 2015) or ComplEx (Trouillon et al., 2016). In this paragraph we briefly describe the relations between these models and our framework, answering if they can still be incorporated in DURENDAL. Any node embedding methods that can be incorporated in an encoder-decoder architecture (Chami et al., 2022) can be also easily incorporated in DURENDAL, following at least one of the two update schemes, by replacing the GNN layers with the methods themselves. We provide a more general view of the DURENDAL framework using the *Update-Then-Aggregate* (UTA) schema in Figure 4. Formally, the equation that describes the forward computation of DURENDAL UTA with any encoder method can be written as:

$$h_{v_t}^{(l)} = \bigoplus_{r \in R} \text{UPDATE}(ENC_\theta^{(l,r)}(h_{v_t}^{(l-1)}), h_{v_{t-1}}^{(l)}) \tag{4}$$

where $ENC_\theta^{(l,r)}$ could be any Graph Representation Learning (Hamilton, 2020) encoder function for node embeddings.

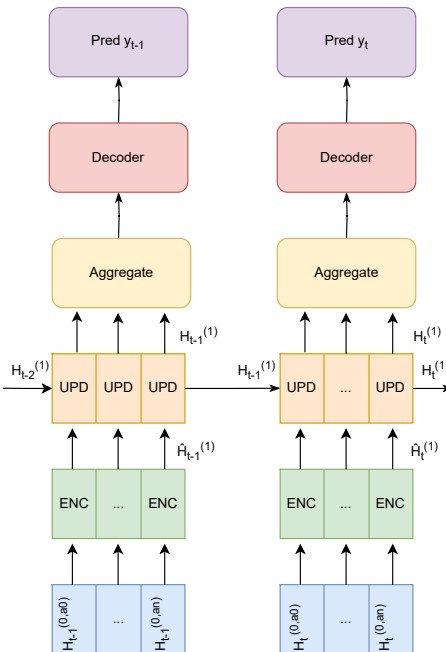

Figure 4: DURENDAL framework with UTA schema and customizable encoder function for node embeddings. ENC could be any Graph Representation Learning (Hamilton, 2020) encoder function for node embeddings, such as GNNs, random-walk, or tensor factorization-based methods.

As an example, if we consider Metapath2Vec, one can train the model on the first snapshot to obtain node embeddings. Then, on the next snapshots, Metapath2Vec learnable parameters can be fine-tuned on the new linearized neighborhoods for nodes to obtain the embedding for the new snapshots, which can be mixed with the historical information of the previous steps using an embedding update module (e.g. GRU). In this case, you can use the *Aggregate-Then-Update scheme*, but not the *Update-Then-Aggregate* one, as Metapath2Vec does not return node embeddings for each relation but a single

node embedding that resumes all the metapaths. To use an *Update-Then-Aggregate* scheme with random-walk-based methods, we may use a Node2Vec encoder (Grover & Leskovec, 2016) for each relation type. However, by using shallow encoders we are not able to generalize on unseen nodes, so DURENDAL loses the inductive property.

**Data-related concerns, gathering and preprocessing.** The data collection campaign for building the `SteemiiTH` dataset has been designed to take into account privacy implications since the information comes from online social networks. In the case of blockchain-based social networks, a first level of pseudo-anonymization is provided by the blockchain itself, however, information within blocks is not encrypted. To minimize the exposure of personally identifiable information we implemented a few on-the-fly strategies which pseudo-anonymize usernames and avoid storing textual content produced by users. Indeed, usernames are hashed and then stored, while for the textual content, we derived semantically meaningful sentence embeddings by Sentence-BERT (SBERT) (Reimers & Gurevych, 2019) and stored the embeddings only. The model has only access to the pseudo-anonymized THN and to text embeddings. The above collection procedure as well as data preprocessing has been approved by the ethics committee of our institution and has been run exclusively on data publicly available through the platform API. Finally, all data is stored within a secure silo. Hence, since we can not release the complete dataset related to `SteemitTH`, we briefly summarize how to collect and preprocess data to obtain it in the following steps:

1. Collect data from June 3, 2016, to February 2, 2017, using the Steemit API (developer documentation, 2021). Consider the first 3 months as the initial training snapshot and the following months as subsequent snapshots. "follow" interactions are available in `custom_json` operations, transactions are available in `transfer` operations, votes in `vote` operations. Posts and comments written by users are available in `comment` operations.

2. Construct a `HeteroData` object with a single node type and four relation types: "follow", "vote", "comment" and "transaction". To construct the edge list related to each relation, you can refer to SteemOps Li et al. (2021a) which describes in detail the schema of each operation and which field contains the ids of the nodes involved in each interaction.

3. For the textual content $X$, we use a pre-trained SBERT language model (Reimers & Gurevych, 2019) to obtain points in the Euclidean space. For each time interval $t$, we call $D_{(u,t)}$ the collection of documents (posts and comments) posted by user $u$ during time interval $t$. To obtain the initial node features $X_{(u,t)}$ of $u$ at time $t$, we average its document embeddings, that is $X_{(u,t)} = \frac{1}{|D_{(u,t)}|} \sum_{d \in D_{(u,t)}} \text{SBERT}(d)$ using the element-wise sum. Users with no published textual content - missing node features - have a zeros vector as initial features.

4. Repeat steps 2-3 for each snapshot.

To process textual content we use the `all-MiniLM-L6-v2` SBERT model. We choose this model because *i*) is trained on all available training data (more than 1 billion training pairs), *ii*) is designed as a general purpose model, and *iii*) is five times faster than the best SBERT model but still offers good quality[7].

For `GDELT18`, `ICEWS18`, and `TaobaoTH`, we download the source data from the PyG library (Fey & Lenssen, 2019). We release the code to compute data preprocessing and obtain the graph snapshot representation to train and test DURENDAL. For further details, you can inspect the annotated code in the GitHub repository[8].

**Societal impacts.** Temporal heterogeneous graph learning benefits a wide range of real-world applications, including but not limited to social network analysis, recommender systems, and research collaboration. However, there are also some potentially negative societal impacts, mainly due to incorrect results returned by an intended usage of the framework. For instance, incorrect predictions on event knowledge graphs may result in false events or relationships between nodes which may stoke misinformation or spread of fake news. Similar considerations hold for the usage of the framework in recommender systems for items or relationship recommendations; in the former incorrect suggestions may produce unfair or even harmful recommendations, while in the latter they may foster different

---

[7]`https://www.sbert.net/docs/pretrained_models.html`, September 2023
[8]`https://anonymous.4open.science/r/durendal-5154/`

kinds of bias and even promote the formation of echo chambers (Ge et al., 2020; Cinus et al., 2022). To mitigate the above outcomes one may act at the recommender system level by adopting different methods to increase fairness, as extensively surveyed in (Wang et al., 2023); while the effects of incorrect event predictions may be mitigated by combining human moderators and validators with DURENDAL to better discern between true and false events.

