# OpenReview forum: "DURENDAL: Graph deep learning framework for temporal heterogeneous networks"
_ICLR.cc/2024/Conference — ICLR 2024 Conference Withdrawn Submission_

### Official Review · Reviewer_fQVE · 2023-10-16

**Soundness:** 3 good
**Presentation:** 1 poor
**Contribution:** 1 poor
**Rating:** 3
**Confidence:** 4

**Summary:**

This paper introduces DURENDAL, a deep learning framework tailored for THNs. DURENDAL adapts to evolving networks and offers two methods to update embeddings. Through testing on new datasets, including one from a Web3 platform and an e-commerce site, DURENDAL proves to be more effective in predictive tasks compared to existing models.

**Strengths:**

1. The authors provide a large number of experiments to analyze the effectiveness of the model.
2. THGs are worth exploring.

**Weaknesses:**

1. The shortcomings of other THNs aren't clarified clearly. For instance, what does *easily incorporate state-of-the-art designs from static GNNs* mean?  And what are the specific drawbacks of these methods? The current presentation lacks clarity, diminishing the paper's motivation when compared to other THNs.  Besides, related work should be cited in the introduction section.
2. This paper's contribution is limited for ICLR standard. The authors primarily employ the ROLAND framework and conventional techniques for heterogeneous graphs. Despite its efficacy, it lacks innovation, potentially falling short of ICLR's acceptance criteria.
3. Recent studies on THNs warrant citation and comparison.
* (1)Fan, Yujie, et al. "Heterogeneous temporal graph neural network." Proceedings of the 2022 SIAM International Conference on Data Mining (SDM). Society for Industrial and Applied Mathematics, 2022.
* (2)Yang, Qiang, et al. "Interpretable Research Interest Shift Detection with Temporal Heterogeneous Graphs." Proceedings of the Sixteenth ACM International Conference on Web Search and Data Mining. 2023.
4. The presentation of this paper is poor.  Additionally, there are typographical errors in the article, such as writing THG as TNH.

**Questions:**

See Weaknesses.

---

> ### Author Response · Authors · 2023-11-17
> **Shortcomings of other Temporal Heterogenous Graph Learning models**
>
> We thank the reviewer for recognizing our contribution to a subfield worth exploring and that we provide a large number of experiments to validate our model. Below we address specific points of the review in detail. We ask the reviewer to increase their score if we address all their main concerns.
>
> > **W1** The shortcomings of other THNs aren't clarified clearly. For instance, what does easily incorporate state-of-the-art designs from static GNNs mean? And what are the specific drawbacks of these methods? The current presentation lacks clarity, diminishing the paper's motivation when compared to other THNs. Besides, related work should be cited in the introduction section.
>
> Temporal heterogeneous networks (THNs) are the datasets on which we perform forecasting tasks. Hence, “other THNs” haven’t shortcomings. Current models that learn from THNs have shortcomings. Regarding temporal graph neural networks (Temporal GNNs), current state-of-the-art approaches are designed for temporal homogeneous graphs only. Factorization-based models (FMs), which represent state-of-the-art models for reasoning over temporal knowledge graphs (the most well-known examples of THNs), are heavily designed to solve link prediction, specifically knowledge graph completion, and cannot be used to make forecasting or any other graph learning task (e.g. node classification), as highlighted in https://openreview.net/pdf?id=Sq9Orta9l5i . For those concerning GNNs for Temporal Heterogeneous Networks, most of the models use a fixed split setting to evaluate the performance of dynamic link prediction (https://dl.acm.org/doi/abs/10.1145/3534678.3539300 ), or do not evaluate it at all (https://dl.acm.org/doi/10.1145/3366423.3380027  see also official comment to reviewer **fYw8). All these statements are already included in the Related Work Section of our paper**. Finally, regarding the meaning of “do not easily incorporate state-of-the-art designs from static GNNs”, considering the following example we hope that they will clarify this aspect. Skip Connections, which are beneficial for GNN architectural design (https://proceedings.mlr.press/v139/xu21k.html ), imply an additional aggregation step for each GNN layer where initial node features are combined with the learned node embeddings. DURENDAL decomposes any heterogeneous GNN layer in two steps: a message-passing operator for each relation and a semantic-level aggregation step. Moreover, it adds a customizable embedding update module before or after any semantic-level aggregation (based on the chosen update scheme). In this way, we allow to include skip connections at every possible aggregation level of a temporal heterogeneous GNN architecture. In our framework, you can easily add a skip connection after the message-passing, and/or after the semantic aggregation, and/or after the update over time. Other temporal heterogeneous GNN models, such as https://dl.acm.org/doi/10.1145/3366423.3380027, https://arxiv.org/abs/2110.13889, or https://aclanthology.org/2020.emnlp-main.462/, treat a GNN as a feature encoder and then build a temporal encoder on top of the GNN (i.e. after the entire GNN architecture, with several GNN layers). Using this model design, you cannot easily incorporate skip connections. Indeed, you can add skip connections only after each semantic aggregation, but not after any possible aggregation function of a THGNN architecture.

---

> ### Author Response · Authors · 2023-11-17
>
> > **W2** This paper's contribution is limited for ICLR standard. The authors primarily employ the ROLAND framework and conventional techniques for heterogeneous graphs. Despite its efficacy, it lacks innovation, potentially falling short of ICLR's acceptance criteria.
>
> The reviewer recognizes that we “provide a large number of experiments to analyze the effectiveness of the model”. As W2, they wrote, “Despite its efficacy, [...]”. Other reviewers, such as **Kepe**, writes “Experimental results look promising on the considered dataset”. From our viewpoint, the main objective of a new proposed solution is to show its effectiveness on the experimental field. A simple solution that works well benefits the machine learning community, especially on a subfield “worth exploring” (as the reviewer recognized as another strength of the work), where there are few or no baselines specifically developed for it.
>
> Moreover, in our paper, we also introduce two new datasets benchmark for temporal heterogeneous networks, which is recognized as a strength by all the other reviewers.
>
> > **W3** Recent studies on THNs warrant citation and comparison.
>
> We thank the reviewer for suggesting these related works.
> However, “Interpretable Research Interest Shift Detection with Temporal Heterogeneous Graphs” is a work on the explainability of shift detection methods for temporal heterogeneous networks, and it is off-topic with respect to temporal heterogeneous graph forecasting, which is the topic of our work.  As written in their work, “We develop a node-type aware and interpretable detection model on temporal heterogeneous graphs. Our model first uses the heterogeneous GNN to learn representations and then constructs sampled sub-graphs as the interpretations“ (see Main contribution in the Introduction). Hence, they utilize heterogeneous GNN, not temporal heterogeneous one, and the temporal information is only utilized to evaluate the shift detection. There are no experiments related to dynamic link prediction as no temporal information is learned.
>
> Regarding the “Heterogeneous temporal graph neural network”, they employ intra and inter-relation aggregation functions between node features to capture the heterogeneous structure of the networks, which are followed by a positional encoding technique to capture temporal information. The model can be easily reconducted in our framework by following the Aggregate-Then-Update schema, where the aggregate function is their intra and inter-relational aggregation, while the update module is represented by the positional encoding technique. However, as mentioned to reviewer **fYw8**, PE cannot be used in the live-update setting. We invite the reviewer to check our official comment to reviewer fYw8. Moreover, we remind the reviewer that our work contains experiments on the effectiveness of our model design (see **Section 5**), showing that similar models to HTGNN, such as HGT, enhance their performance if repurposed in our framework.
>
> > **W4** The presentation of this paper is poor. Additionally, there are typographical errors in the article, such as writing THG as TNH.
>
> We ask the reviewer to improve the quality of their review explaining why s/he finds the presentation of this paper poor. The reviewer reports just one typographical error in the work, which we believe is not sufficient to assign a score of one. In addition, reviewer **qqSc** finds the presentation of our work excellent, while reviewer **Kepe** says that “The paper is overall well written, easy to follow and with good references for people that might be approaching the field of dynamic graphs for the first time”.

---

> > ### Comment · Reviewer_fQVE · 2023-11-21
> >
> > Thanks for the authors' feedback. I agree with the contribution of introducing two new temporary heterogeneous graphs datasets, which is beneficial for data mining community.  However, the paper's proposed model is a straightforward extension of ROLAND, specifically tailored for discrete dynamic graphs rather than consistent dynamic graphs, which limits the insight offered by this paper.  Thus, I maintain that the contribution of this paper is limited for ICLR. It appears to be more appropriately suited for data mining community, given it's extension on dynamic GNN and two new datasets.

---

> > > ### Author Response · Authors · 2023-11-22
> > >
> > > Thanks for the reviewer's comment. We all agree that the work is incremental w.r.t. ROLAND but we do not agree that being an incremental work is, in absolute, a weakness, especially in a field THG which you consider "worth exploring". From this perspective, the impression is that there is a discrepancy between the overall quantitative evaluation and the last comment: 3 as score is for papers showing technical lacks, and according to most of the reviewers who have replied to the rebuttal this is not the case. So, even maintaining an overall negative judgment of the paper, you should think about increasing the score to 4.
> > > Moreover, it is quite obscure what "consistent" means for dynamic graphs. Maybe, you could clarify that.

---

### Official Review · Reviewer_Kepe · 2023-10-27

**Soundness:** 2 fair
**Presentation:** 3 good
**Contribution:** 2 fair
**Rating:** 6
**Confidence:** 4

**Summary:**

The paper proposes an extension of ROLAND for discrete-time temporal heterogeneous graphs (where the temporal graph is described as a sequence of snapshots), together with two new datasets that can be used for evaluation. The main methodological novelty of the paper is how to incorporate an aggregation mechanism across various edge types through two possible different schemes: update-then-aggregate and aggregate-then-update. For what concerns the datasets, two new temporal heterogeneous graphs are introduced in the paper: TaobaoTH (a dataset of user behaviour provided from Taobao - an online shopping platform) and SteemitTH (a dataset of user interactions from Steemit - a blockchain-based social network). The proposed model is evaluated on multi-relation and mono-relational link prediction on the proposed datasets plus two other datasets that have already been used in the literature (GDELT18, ICEWS18). The approach appears to perform well on the considered datasets when compared to 9 selected baselines.

**Strengths:**

The paper is overall well written, easy to follow and with good references for people that might be approaching the field of dynamic graphs for the first time. While the approach appears rather straightforward (especially when compared to ROLAND), experimental results look promising on the considered dataset. The introduction of new datasets is also something that the community will most likely benefit from.

**Weaknesses:**

As it might have emerged from my comments in the “Strengths” section, the approach appears to be a not particularly original improvement over ROLAND (unless my understanding is wrong, the main addition is the introduction of an aggregation mechanism across multiple relations and the use of heterogeneous GNNs for feature extrapolation). On top of this, while yes the method appears to show good results on the considered datasets compared to the baselines, I’ve some doubts about the experimental evaluation. In particular, have the baselines considered in the experiment been tuned for the dataset? Taking for instance TGN from Rossi et al, the model was not evaluated on any of the datasets used in the paper. As such, if such architecture was not tuned (as instead the proposed approach was), we might be observing lower performance for such methods (as well as the other baselines), which are simply due to a suboptimal architectural choice. I’d greatly appreciate if the authors could comment on this in their rebuttal

**Questions:**

Besides what highlighted above, I have a few questions / comments that I would the authors to address:

1) Many methods appear to achieve on TaobaoTH a PR AUC that is consistent with random guessing in a balanced binary classification problem. This suggests that many models are actually not learning anything meaningful on that dataset. Can you please clarify why this might be the case?

2) The fact that TGN and CAW do not compute in their implementation the MRR, I believe it’s not a good reason to avoid computing such statistics for these methods. I’d encourage the authors to fix the implementation in this case to provide a better comparison of all methods.

3) I’m confused why many MRRs appear equal to 0.5, can the authors provide some details on the implementation they used for this and how negatives were sampled?

4) In section 4 it is stated that “minimum number of snapshots to allow live-update evaluation” is four, can you provide some details on why that is the case? From algorithm 2 in ROLAND my understanding is that 2 steps are enough for live update evaluation

---

> ### Author Response · Authors · 2023-11-17
> **Official Comment (Weaknesses)**
>
> We thank the reviewer for their feedback and for finding that our work is overall well written, easy to follow, and with good references for people who might be approaching the field of dynamic graphs for the first time. Below we address specific points raised in the review in detail. We ask the reviewer to increase their score if we address all their concerns.
>
> ### Weaknesses
>
> As stated in our work, the two main methodological innovations are i) proposing a novel graph deep learning framework that allows an easy repurposing of any heterogenous GNNs to a dynamic setting; hence, the introduction of an aggregation mechanism across multiple relations and the use of heterogeneous GNNs for feature extrapolation **are not the main addition of our work** but the way we decompose any heterogeneous GNN layer to allow its extension in a dynamic setting. ii) the introduction of two different update schemes, namely Upgrade-Then-Aggregate and Aggregate-Then-Update, for obtaining temporal heterogeneous node embeddings, highlighting their strengths and weaknesses and their practical use scenarios. The straightforward way to use ROLAND on heterogeneous networks would be just adding an update mechanism after each heterogeneous GNN layer. In our work,  instead, we propose two new different update schemas that are different from the ROLAND one and are more suitable for heterogeneous graph learning as they allow to capturing relational temporal dynamics (see **Section 3**). We invite also the reviewer to see the paragraph on “effectiveness of update schemes” in **Section 5**, where we compare our update schemes with the ROLAND one, showing an improvement for both schemes.
>
> On top of this, We believe **hyperparameter-tuning all the baselines is a crucial task** for the experimental evaluation of graph deep-learning models as not tuning existing models on new datasets leads to an unfair evaluation of the new proposed ones, as unfortunately highlighted in very few works on GNNs (https://openreview.net/forum?id=HygDF6NFPB). We invite the reviewer to read the **original Appendix** of our work to find all the information about hyperparameter tuning for all the models, including baselines. For the two continuous-time models, TGN and CAW, as highlighted in our paper, we believe their continuous-time representation for temporal networks is not beneficial in application scenarios where datasets are snapshots-based. Indeed, for instance, TGN works typically with a list of timed events that happen one before another, while in our datasets there are large batches of events (i.e. appearance of new links) that happen all at the same time (i.e. in snapshots). Hence, we believe the problem for TGN is **architectural rather than related to tuning** and its core mechanism should be revised to work with snapshot-based datasets. However, we conducted a **hyperparameter search for TGN** using the following grid search: learning rate {0.1, 0.01, 0.001, 0.0001, 0.00001}, embedding_dim {50, 100, 200, 300, 400, 500}, memory_dim {50, 100, 200, 300, 400, 500}. The results show the best architecture is the one reported in the paper, which uses lr equal to 1e-5, and embedding_dim = memory_\dim = 100.
>
> | embedding_dim / learning_rate | 0.1 | 0.01 | 0.001 | 0.0001 | 0.00001 |
> | ----------------------------- | --- | ---- | ----- | ------ | ------- |
> | 50                            | 0.5 | 0.5  | 0.880 | 0.888  | 0.885   |
> | 100                           | 0.5 | 0.5  | 0.878 | 0.886  | **0.889**   |
> | 200                           | 0.5 | 0.5  | 0.770 | **0.889**  | **0.889**   |
> | 300                           | 0.5 | 0.5  | 0.811 | 0.860  | 0.866   |
> | 400                           | 0.5 | 0.5  | 0.810 | 0.828  | 0.881   |
> | 500                           | 0.5 | 0.5  | 0.821 | 0.832  | 0.883   |

---

> > ### Author Response · Authors · 2023-11-17
> > **Official Comment (Questions)**
> >
> > > **Q1** Many methods appear to achieve on TaobaoTH a PR AUC that is consistent with random guessing in a balanced binary classification problem. Can you please clarify why this might be the case?
> >
> > As written in our work, “On TaobaoTH, we obtain surprising results. The best performance is achieved by HAN, that do not use leverage temporal information, apart from the incremental training. TGN and CAW achieve notably worse prediction performance than heterogeneous GNNs, while EvolveGCN, GCRN-GRU, and HetEvolveGCN obtain poor performance. DURENDAL reaches good performance using an embedding update module that simply computes a convex combination between the past and the current representation of nodes, with a past coefficient no greater than $0.1$. The same results are obtained using a time granularity of one or ten minutes. Hence, predicting future ``buy'' relations seems just related to the other actions performed by users on items (view an item, add it to your favorites or in your cart) in the previous snapshot, not to the order they are carried out, nor to repetition over time. The result is surprising because sophisticated dynamic models seem to give too much importance to past information without learning this simple structural pattern. However, it is important to note that {\tt TaobaoTH} has a very low evolutivity value, equal to $0.003$” (see **Section 5**). We invite the reviewer to carefully read the official comment to reviewer qqSC, where we clarify some aspects on the construction of this dataset. Consequently, snaphot-based homogeneous models seems to not be able to capture the entire process of purchasing an item for an active user spanning over one day of e-commerce session.
> >
> > > **Q2** The fact that TGN and CAW do not compute in their implementation the MRR, I believe it’s not a good reason to avoid computing such statistics for these methods. I’d encourage the authors to fix the implementation in this case to provide a better comparison of all methods.
> >
> > We are conscious that providing MRR also for TGN and CAW provides a better comparison of all methods. However, fixing their official implementation to compute MRR is not so straightforward (i.e. it is not just a matter of importing a sklearn metric). In the Temporal Graph Learning literature, we can find several works where some evaluation metrics on datasets are not computed if the original implementation does not provide it (To cite a few: https://ojs.aaai.org/index.php/AAAI/article/view/20746, https://ojs.aaai.org/index.php/AAAI/article/view/16802/16609, https://aclanthology.org/2020.emnlp-main.462/ )
> >
> > > **Q3** I’m confused why many MRRs appear equal to 0.5, can the authors provide some details on the implementation they used for this and how negatives were sampled?
> >
> > We compute MRR using its definition: the average reciprocal rank of the scores of positive edges against the score of negative edges. Negative edges are obtained by corrupting the positive edges using randomly sampled destinations. As stated in our paper, we sampled negative edges for scalability issues. Random negative sampling is a standard practice in the literature (https://ojs.aaai.org/index.php/AAAI/article/view/5984/5840 ). We sampled one negative edge for each positive edge. We discuss the limitations this choice implies in the **original Appendix** of our work.
> >
> > > **Q4** In section 4 it is stated that “minimum number of snapshots to allow live-update evaluation” is four, can you provide some details on why that is the case? From algorithm 2 in ROLAND my understanding is that 2 steps are enough for live update evaluation.
> >
> > We thank the reviewer for asking this clarification. From Algorithm 2 in ROLAND, 2 steps are enough for computing live update evaluation: the model is trained on the first snapshot and tested on the second. However, in practice, to completely leverage the live-update protocol both in terms of training and evaluation strategies, you need at least three snapshots: the model is trained on the first and tested on the second, then is fine-tuned on the second and tested on the third. Using just two snapshots is equivalent to a fixed split setting, which does not take into account the evolving nature of the data. Lastly, we require at least four snapshots as there are some fixed split settings in which the validation set is constructed as a future time snapshot (https://arxiv.org/abs/2307.01026, https://openreview.net/forum?id=1GVpwr2Tfdg ). In these scenarios, we need four snapshots: the first to train the model, the second for the dev set, the third to test the model, and then the fourth to re-test the model after fine-tuning. Hence, we require a minimum number of four snapshots to allow a  complete general run of the live-update protocol for training and evaluating the models. We plan to clarify this aspect by integrating this answer into the paper.

---

> > > ### Comment · Reviewer_Kepe · 2023-11-20
> > >
> > > I thank the reviewers for their response. I have a couple of follow-up comments based on their rebuttal:
> > >
> > > 1. Thank you for highlighting that the baselines have indeed been tuned, can you please confirm that the tuning has been done per dataset and not only on a reference one (e.g. SteemitTH) and then the same architecture re-applied across all datasets?
> > >
> > > 2. For what concerns Q2, I still believe that not reporting MRR for some architecture is suboptimal (this is irrespective of what some prior work might have done), and it would have been better to have such metric listed for all approaches

---

> > > > ### Author Response · Authors · 2023-11-20
> > > >
> > > > Thank you for your response.
> > > >
> > > > 1 - Sure, in the rebuttal we reported the performance of TGN using the specified grid search on SteemitTH just to show an example for our statement on continuous-time dynamic graph models. Additionally, we chose TGN as an example because it was cited in your review, we fine-tuned all the models with the grids reported in the Appendix for all the datasets.
> > > >
> > > > 2- We understand your concern and we agree with it. We are working to modify the original implementations of TGN and CAW to compute MRR and report their results using this evaluation metric. However, due to the computational cost of their learning process, it will be very difficult to show the results within the rebuttal deadline. We plan to add MRR results for TGN and CAW in a future version of the work.
> > > >
> > > > If you have any additional questions or concerns, please feel free to let us know. We are open to further discussion and would be happy to provide any additional clarification needed. If there are no other concerns, we kindly request to increase your score.
> > > >
> > > > Thank you for your time and valuable feedback.

---

> > > > > ### Comment · Reviewer_Kepe · 2023-11-21
> > > > >
> > > > > Thank you for your response. In general I find this paper to be very borderline. I tend to agree with reviewer fQVE that the contribution is not particularly novel if compared to ROLAND. However, given the generally good results, in light of the rebuttal I'm willing to raise my score to a 6

---

### Official Review · Reviewer_qqSc · 2023-10-31

**Soundness:** 3 good
**Presentation:** 4 excellent
**Contribution:** 3 good
**Rating:** 6
**Confidence:** 5

**Summary:**

This work proposes a generic framework of adapting static heterogeneous GNNs to the dynamic setting through two types of schemes: Update-then-Aggregate (UTA) and Aggregate-then-Update (ATU). The authors also introduce two new datasets of dynamic heterogeneous graphs (TaobaoTH and SteemitTH) for future benchmarking. The proposed method achieves better performance in future link prediction tasks on all four datasets.

**Strengths:**

1. The designed framework is generic and can be integrated with any static heterogeneous GNNs. Given its simplicity and wide adaptivity, it can facilitates future research on dynamic heterogeneous graph learning.
2. This work introduces two new benchmark datasets of dynamic heterogeneous graphs, including one dataset from e-commerce recommendation and one dataset from blockchain-based online social network. Specifically, the TaobaoTH is of a relatively large size with ~360k nodes.
3. The designed method achieves a better performance compared to the existing baselines including static GNNs, and dynamic GNNs.

**Weaknesses:**

1. Based on my understanding of the differences between dynamic graphs and temporal graphs, I think it would be better if this work is positioned for dynamic heterogeneous networks instead of temporal heterogeneous networks. Dynamic networks are snapshot-based networks, i.e., aggregating edges and nodes within certain time windows, which is exactly what this paper considers. In contrast, temporal networks are more dynamically changing where each edge is associated with a timestamp (not a snapshot).
2. It is not clear what scheme for the proposed method is applied in Table 2.

**Questions:**

1. What are the differences or new aspects between the existing Taobao benchmark (https://pytorch-geometric.readthedocs.io/en/latest/generated/torch_geometric.datasets.Taobao.html#torch_geometric.datasets.Taobao) and the one introduced by this paper?

2. The number of edges of TaobaoTH is even smaller than the number of nodes. Can you elaborate why this graph is so sparse?

3. The evolutivity of TaobaoTH is extremely low. Does it mean there are very few new edges across snapshots? Or is it because at different snapshots, edges are repetitive (e.g., user viewed an item at snapshot-1 and viewed the same item at snapshot-2)? On this question, I think it's also worth reporting the repetitive metrics of the datasets.

---

> ### Author Response · Authors · 2023-11-17
>
> We thank the reviewer for their feedback and for finding that our work facilitates future research on dynamic heterogeneous graph learning. Below we address specific points raised in the review in detail. We ask the reviewer to increase their score if we address all their concerns.
>
> > **W1** Based on my understanding of the differences between dynamic graphs and temporal graphs, I think it would be better if this work is positioned for dynamic heterogeneous networks instead of temporal heterogeneous networks.
>
> Based on the literature (https://dl.acm.org/doi/abs/10.5555/3455716.3455786), the description provided by the reviewer refers to the distinction between discrete-time and continuous-time dynamic graphs. Temporal and dynamic, instead, are typically used as synonyms (https://dl.acm.org/doi/10.1145/3534678.3539300). There are also examples in the literature where the term dynamic is used for continuous-time graphs ( https://openreview.net/forum?id=1GVpwr2Tfdg). In this sense, the same ambiguity is also present in network science where temporal networks and dynamic networks are often interchanged. Also from the network science perspective we prefer referring to temporal networks since dynamic networks are more related to dynamic models, while the former include also snapshot-based representation (for instance https://link.springer.com/book/10.1007/978-3-030-23495-9 )
>
> > **W2** It is not clear what scheme for the proposed method is applied in Table 2.
>
> We thank the reviewer for exposing this doubt. Table 2 reports the best DURENDAL results across both update schemes. Specifically, the best results for GDELT18, ICEWS18, TaobaoTH, and SteemitTH are obtained using UTA, ATU, ATU, and UTA, respectively. We plan to add a Table on update schemes in the Appendix.
>
> ### Questions
> We thank the reviewer for the insightful questions on the TaobaoTh dataset that helped us to clarify some aspects of the construction of this benchmark. We plan to integrate the three following answers to the content of our work. We remind to reviewer that **the code of our work is available** in an anonymized Github Repository provided in the pdf. The repository contains the code to construct TaobaoTH starting from the PyG version.
>
> > **Q1** What are the differences or new aspects between the existing Taobao benchmark (https://pytorch-geometric.readthedocs.io/en/latest/generated/torch_geometric.datasets.Taobao.html#torch_geometric.datasets.Taobao) and the one introduced by this paper?
>
> The original benchmark dataset for Taobao was a dataset of user behaviors used in prior works on recommendation systems and provided by the Tianchi Alicloud platform https://tianchi.aliyun.com/dataset/649. Recently, as indicated by the reviewer, has been added as a temporal heterogeneous network in PyTorch Geometric, one of the most well-known and used libraries for graphML. As stated in our paper, “we construct heterogeneous graph snapshots with time granularity equal to five minutes. We consider a heterogeneous subgraph induced by 250k random sampled items for scalability issues”(See Section 4). Hence, our dataset is a subgraph induced by 250k random sampled items of the PyG dataset aggregated in snapshots with a time granularity of five minutes. We obtained 288 snapshots considering the first day of user-item interactions.
>
> > **Q2** The number of edges of TaobaoTH is even smaller than the number of nodes. Can you elaborate why this graph is so sparse?
>
> The experimental evaluation is conducted in a transductive setting. For TaobaoTH, the users considered in the dataset are the nodes active (i.e. they have at least interacted with one item) on the first five minutes of the dataset (from t=0 to t=300). Hence, the datasets consider all the relations between active users on the first session, which represent the seen nodes during training, and 250k random sampled items. This choice led to a very sparse dataset compared to the original PyG but it allowed us to study simple temporal patterns between the relation “pageview”, “add to cart”, and “buy” within e-commerce sessions in a day, as described in **Section 5**.
>
> > **Q3** The evolutivity of TaobaoTH is extremely low. [...]
>
> As defined in the paper, “evolutivity is the average number of new links in the snapshots (i.e. $\frac{1}{|T-1|}\sum_{t=1}^{T} |E_t|$). This definition implies that new edges are defined as edges with a future timestamp w.r.t the current one; hence, an extremely low evolutivity means there are few new edges across snapshots. In our intuition, this is reasonable as the dataset follows the entire buying process in a day of the users active in the first 5 minutes of the session. For repetitive metrics, we compute the average cardinality of the intersection between the first snapshot and the others for each relation type. Only about 7% of pageviews are repetitive, while all the interactions with the cart and the purchases are completely new.

---

### Official Review · Reviewer_fYw8 · 2023-11-01

**Soundness:** 2 fair
**Presentation:** 1 poor
**Contribution:** 2 fair
**Rating:** 3
**Confidence:** 3

**Summary:**

This paper proposes DURENDAL, a training framework for temporal heterogeneous networks. It introduced two training schemes, Update-Then-Aggregate and Aggregate-Then-Update, which are different aggregation methods for training. It then benchmarks the performance on four datasets.

**Strengths:**

1. Benchmarking dynamic heterogeneous graphs is important.
2. Two datasets are introduced by transforming the original open datasets.
3. Experiments on performed.

**Weaknesses:**

1. The mechanism of why DURENDAL outperforms baselines is unclear.
2. The comparison of UTA and ATU is not clear. System-level (e.g. run time, memory usage) evaluation might be helpful.
3. More commonly used datasets are needed if the paper wants to be a benchmark paper (e.g. Open Academic Graph).

**Questions:**

1. Where is the figure for Aggregate-Then-Update (ATU)?
2. Why does DURENDAL have better accuracy than baselines?
3. Why does not the paper compare with [1]?

[1] Hu, et al. "Heterogeneous graph transformer." Proceedings of the web conference 2020.

---

> ### Author Response · Authors · 2023-11-17
>
> We thank the reviewer for their questions. Before addressing the specific points raised in the review, we ask the reviewer what “Experiments on performed” (Strengths 3) means. Below we address specific points raised in the review in detail. We ask the reviewer to increase their score if we address all its concerns.
>
> > **W1** The mechanism of why DURENDAL outperforms baselines is unclear. **Q2** Why does DURENDAL have better accuracy than baselines?
>
> DURENDAL is a general framework for temporal heterogeneous networks. It introduces hierarchical heterogeneous node states as node embedding, allowing the repurposing of heterogeneous GNNs to a dynamic setting. Moreover, it introduces two new update schemas for updating heterogeneous node embedding over time, which are based on modeling relational temporal dynamics (i.e. different temporal patterns for each specific type of relation). The latter represents an innovation in temporal graph learning models, as typically the temporal encoder is placed only before or after a heterogenous feature encoder (GNN of Factorization-based), without a strictly intertwine between the heterogeneity and dynamicity aspects (https://dl.acm.org/doi/10.1145/3366423.3380027, https://arxiv.org/abs/2110.13889 ). We remind the reviewer that our two newly introduced update schemes outperform the ROLAND one (see “effectiveness of update schemes”). To clearly understand what relation temporal dynamics are and potential use cases for the two update schemes, we invite the reviewer to read the Appendix of our work.
>
> Below we explain, in our intuition, why DURENDAL outperforms each of the other baselines:
> - GAT, HAN, ComplEx: They are static models so they do not capture the temporal information, which is crucial to understanding the network evolution, and predicting future links. We invite the reviewer to read also the paragraph “Effectiveness of model-design”, where we repurpose HAN and other well-known heterogenous GNNs in our framework, showing an improvement in their performance.
> - TGN, CAW: They are continuous-time dynamic graph models for future link prediction. As stated in our paper, “ their continuous-time representation for temporal networks is not beneficial in application scenarios where datasets are snapshots-based”, as in our case. It is worth noting that discrete-time dynamic graphs in live update settings represent an appealing scenario for many real-world applications (https://dl.acm.org/doi/abs/10.1145/3534678.3539300, https://arxiv.org/abs/2307.01026)
> - EvolveGCN, GCRN-GRU, TNTComplEx: the first two models are discrete-time dynamic graphs models, the latter is a strong factorization-based model baseline for temporal knowledge graphs, which are typically collected by snapshots. DURENDAL achieves better performance than these baselines because they do not model relational temporal dynamics. Hence, they do not model in different ways the evolution of each relation; instead, they use a single strategy to model the overall graph dynamics.
> - HetEvolveGCN: it is a baseline we developed to show the effectiveness of our solution. It is based on EvolveGCN and uses an EvolveGCN model for each relation, combining the generated node embeddings through a semantic aggregation function. Hence, it is a snapshot-based architecture that also models relational temporal dynamics. DURENDAL achieves better performance because it captures relational temporal dynamics in an effective way. Indeed, EvolveGCN uses a recurrent neural architecture to update only the learnable parameters of the model, while DURENDAL is based on embedding evolution, updating directly the generated node embeddings (https://arxiv.org/abs/2302.01018).

---

> > ### Author Response · Authors · 2023-11-17
> >
> > > **W2** The comparison of UTA and ATU is not clear. System-level (e.g. run time, memory usage) evaluation might be helpful.
> >
> > We report the computational complexity (provided in the original Appendix), number of learnable parameters (to obtain byte memory usage you can just multiply these numbers by  4, that is the memory usage of a single TensorFloat parameter, and run-time of both update schemes (UTA and ATU) in the following table. We plan to add the resume table in the Appendix of our paper. UTA has a higher number of learnable parameters compared to ATU as it uses embedding update modules for each relation in the dataset, while ATU uses a single update module for all the relations. For the running time on the training loop, UTA scales well and typically less than linearly w.r.t. the number of parameters. The computational complexity for UTA and ATU is k * (O(|E|) + |R| * O(update)), and k * (O(|E|) + O(update)), respectively, where $k$ is the number of layers, O(update) is the computational complexity of the chosen embedding update modules (e.g. a GRU).
> >
> >  ~                                | UTA                            | ATU
> > ----------------------------------|--------------------------------|---------------------------
> >  SteemitTH \#parameters           | 3.111.426                      | 1.629.954
> >  SteemitTH running time (seconds) | 892,32                         | 677.60
> >  ICEWS18 \#parameters             | 9.405.698                      | 1.905.410
> >  ICEWS18 running time             | 1390,76                        | 302,86
> >  GDELT18 \#parameters             | 9.405.698                      | 1.905.410
> >  GDELT18 run time                 | 129,50                         | 36,42
> >  TaobaoTH \#parameters            | 1.929.994                      | 1.929.986
> >  TaobaoTH running time            | 27.546,12                      | 27.546,04
> >
> > > **W3** More commonly used datasets are needed if the paper wants to be a benchmark paper (e.g. Open Academic Graph).
> >
> > We thank the reviewer for suggesting another potential data source for gathering a temporal heterogeneous network dataset. The main objective of our work is to propose a deep learning framework for THNs, arising from a combination of model design principles from temporal and heterogeneous graph learning and introducing two novel update schemes based on relational temporal dynamics. We conducted an extensive evaluation of our model on four different datasets against nine different baselines to show its prediction power. However, this work is not a benchmark-oriented paper, and, as reviewer fQVE said, “The authors provide a large number of experiments to analyze the effectiveness of the model”.
> >
> > > **Q1** Where is the figure for Aggregate-Then-Update (ATU)?
> >
> > We thank the reviewer for suggesting adding the figure for the ATU update scheme. We plan to add it to the Appendix.
> >
> > > **Q3** Why does not the paper compare with [1]?  [1] Hu, et al. "Heterogeneous graph transformer." Proceedings of the Web Conference 2020.
> >
> > Heterogeneous Graph Transformer (HGT) design node- and edge-type dependent parameters to characterize the heterogeneous attention over each edge, empowering HGT to maintain dedicated representations for different types of nodes and edges. Its architecture can be easily repurposed in DURENDAL for temporal heterogeneous network forecasting. Indeed, we’ve already repurposed HGT in our framework, showing an improvement in its performance by our model design. Hence, we invite the reviewer to carefully read the original paragraph “Effectiveness of model design” in Section 5 of our work. In the original WWW work, the authors introduce the relative temporal encoding technique based on positional encoding into HGT to handle dynamic aspects. However, they do not evaluate the performance of their architecture on temporal heterogeneous networks and they use it just to show a case study(see Section 5.4 of their paper). Moreover, their technique cannot be used in the live-update setting, as a) it requires that the model have access to all the snapshots at the same time, while in the live-update setting, the model has access just to one snapshot at a time, leading to a more tricky testing ground for temporal graph learning models; b) Their relative temporal encoding technique works with timestamp on nodes only (see Section 3.5 of their paper). In the transductive live-update setting, all the nodes appear in the first snapshots, so all the time gaps would be zero; timestamps refer to edges.

---

### Author Response · Authors · 2023-11-17
**Thank you**

We would like to thank the reviewers for their helpful comments and actionable suggestions for improving our manuscript. We are delighted that the reviewers find that our work “facilitates future research on dynamic heterogeneous graph learning” (**qqSc**), is “well written, easy to follow and with good references for people that might be approaching the field of dynamic graphs for the first time” (**Kepe**). Furthermore, we appreciate that all the reviewers agree on the benefits of the introduction of two new datasets as well as that we “provide a large number of experiments to analyze the effectiveness of the model” (**fQVE**).
We thank also the reviewers for the insightful questions that will help improve the clarity of our discussion. We responded to each review directly to address their specific comments. We welcome further comments or requests during the discussion period. In the rebuttal, sometimes we invited the reviewer to **read the original Appendix of our works**, where some of their concerns are already addressed. We hope that the reviewers revise their scores in light of our response.